# Population Genomics Study and Implications for the Conservation of *Zabelia tyaihyonii* Based on Genotyping-By-Sequencing

**DOI:** 10.3390/plants12010171

**Published:** 2022-12-30

**Authors:** Halam Kang, Sung-Mo An, Yoo-Jung Park, Yoo-Bin Lee, Jung-Hyun Lee, Kyeong-Sik Cheon, Kyung-Ah Kim

**Affiliations:** 1Department of Biological Science, Sangji University, Wonju 26339, Republic of Korea; 2Department of Biology Education, Chonnam National University, Gwangju 61186, Republic of Korea; 3Environmental Research Institute, Kangwon National University, Chuncheon 24341, Republic of Korea

**Keywords:** Caprifoliaceae, genetic diversity, population structure, Taihyun’s abelia, endemic species, rare plant, single nucleotide polymorphism

## Abstract

*Zabelia tyaihyonii* (Nakai) Hisauti and H. Hara is a perennial shrub endemic to Republic of Korea that grows naturally in only a very limited region of the dolomite areas of Gangwon-do and Chungcheongbuk-do Provinces in the Republic of Korea. Given its geographical characteristics, it is more vulnerable than more widely distributed species. Despite the need for comprehensive information to support conservation, population genetic information for this species is very scarce. In this study, we analyzed the genetic diversity and population structure of 94 individuals from six populations of *Z. tyaihyonii* using a genotyping-by-sequencing (GBS) approach to provide important information for proper conservation and management. Our results, based on 3088 single nucleotide polymorphisms (SNPs), showed a mean expected heterozygosity (*He*) of 0.233, no sign of within-population inbreeding (*G_IS_* that was close to or even below zero in all populations), and a high level of genetic differentiation (*F_ST_* = 0.170). Analysis of molecular variance (AMOVA) indicated that the principal molecular variance existed within populations (84.5%) rather than among populations (17.0%). We suggested that six management units were proposed for conservation considering Bayesian structure analysis and phylogenetic analysis, and given the various current situations faced by *Z. tyaihyonii*, it is believed that not only the in situ conservation but also the ex situ conservation should be considered.

## 1. Introduction

*Zabelia tyaihyonii* (Nakai) Hirsute and H. Hara (Taihyun’s abelia) (Figure 1A) belongs to Caprifoliaceae Juss., and is a deciduous broad-leaved shrub with a height of approximately 1–2 m. This species was first described as a species of the genus *Abelia* by Nakai in 1921 [1]. However, Hisauchi and Hara [2] separated *Zabelia* (Rehder) Makino from *Abelia* into a new genus based on the external morphological and anatomical characteristics suggested in a previous study [3], so this is generally treated as the current scientific name.

This species, first described as a species of the genus *Abelia* by Nakai in 1921 [1], has this current scientific name because Hisauchi and Hara [2] segregated *Zabelia* of the genus *Abelia* into a new genus, *Zabelia* (Rehder) Makino, based on the external morphological and anatomical characteristics suggested in a previous study [3]. This species is very similar to *Zabelia mosanensis* (T.H. Chung ex Nakai) Hisauti and Hara, but it is distinguished in that its leaf length is half or less, and hair on the filament is absent [4]. In the current classification system, however, it is common to treat the two taxa as one taxon and treat *Z. mosanensis* as a synonym for *Z. tyaihyonii* according to the principle of priority [5].

This species is a very important resource because it is a Korean endemic plant and an indicator species of limestone areas [6]. It is also listed as a rare plant [7,8], and V Grade of the Floristic Target Species (FT species) in the Republic of Korea [9] because it grows naturally in only a very limited region of the dolomite areas of Gangwon-do and Chungcheongbuk-do Provinces in the Republic of Korea [9]. Additionally, *Z. tyaihyonii* has very high value as garden material because it has gorgeous and fragrant flowers [10].

Since endemic plants are very sensitive to subtle changes in environmental factors, they should be prioritized for management and preservation [11,12]. Species with a very narrow range, such as *Z. tyaihyonii*, are particularly vulnerable. It is important to establish a conservation strategy based on ecological conditions and the genetic diversity suitable for each species to prevent extinction and maintain continuous natural populations [13,14,15]. Previous studies on *Z. tyaihyonii* were mainly focused on systematics [16,17,18,19,20,21], and only four studies have been performed on its conservation biology [4,10,22,23]. The ecological characteristics of *Z. tyaihyonii* habitats were very thoroughly described in three studies conducted on all native habitats [4,10,23]. However, information on genetic diversity is very scarce; only one previous study considered this topic and it included only two natural populations [22].

Next-generation sequencing (NGS) technology has transformed modern biology, including population genetics, due to its high throughput and low cost [23,24]. The two strategies for NGS include whole-genome resequencing (WGR) and reduced-representation sequencing (RRS) [24,25]. Genotyping-by-sequencing (GBS) is one of most widely used RRS methods in which the barcode system is improved to allow the discovery of genome-wide single nucleotide polymorphisms (SNPs) with a lower error rate and low cost [26]. Because of these advantages, the high-throughput SNPs detected by GBS are widely used for genetic diversity analysis in many plant species [24,27,28,29,30,31,32,33,34].

In this study, GBS was used to genotype 94 *Z. tyaihyonii* individuals collected from all known natural habitats (six populations) (Figure 1B). The main goals of this study were to: (1) Assess the genetic diversity and population genetic structure of this species; (2) Develop and implement the conservation strategies based on genetic diversity. The results obtained from this study will provide fundamental baseline knowledge for developing and implementing conservation strategies.

## 2. Results

### 2.1. SNP Analysis and Genetic Diversity

An initial quality check of 663,388,244 reads after sequencing resulted in a matrix of 48,326 SNPs. Initial filtering (i.e., missing SNP data, minor allele count, minimum quality score, and minimum read depth) resulted in the identification of 31,147 SNPs. A further filtering step (individuals with missing data) resulted in a total of 4085 SNPs, and the final filtering steps yielded 3088 final SNP loci in 94 genotyped individuals across six sampled locations.

### 2.2. Genetic Diversity

Genetic diversity indices within the six populations of *Z. tyaihyonii* were estimated and are summarized in Table 1. The means for the number of different alleles (*Na*) and number of effective alleles (*Ne*) were 1.734 and 1.389, respectively. Shannon’s information index (*I*) was the highest in BS (0.406) and the lowest in SH (0.249), with an average of 0.355. The ranges of observed heterozygosity and expected heterozygosity among the six populations were from 0.209 (SH) to 0.273 (TG) and from 0.170 (SH) to 0.263 (BS), respectively. Additionally, inbreeding coefficient (*G_IS_*) was negative in all populations except BS.

### 2.3. Genetic Differentiation and Population Structure

The results from the analysis of molecular variance (AMOVA) (Table 2) revealed that only 17.0% of the total generic variation was due to among-population differentiation and that the majority of differentiation was within individuals (84.50%). The *F-statistics* from the analysis also showed that the species discussed in this study was moderately to highly structured, with a value significantly different from zero after 20,000 permutations (*F_ST_* = 0.170, *p* < 0.001). Although there was a significant expected reduction in heterozygosity among all individuals (*F_IT_* = 0.155, *p* < 0.001), the slightly negative total inbreeding coefficient (*F_IS_* = −0.018) was not shown to significantly differ from zero (*p* > 0.05), indicating little or no inbreeding in the species.

The coefficient of genetic differentiation (*F_ST_*) and a high level of historical gene flow (*N_m_*) of pairs of populations were detected in *Z. tyaihyonii* (Appendix A). Population pairs including SH exhibited higher *F_ST_* and lower *N_m_* values than other pairs, while the lowest genetic differentiation (*F_ST_* = 0.072) and the highest gene flow (*N_m_* = 3.217) were observed between BS and CW (Appendix A). The positions of inferred gene flow barriers between populations were identified based upon the matrix of *F_ST_* values (Figure 1).

All estimator criteria, median of means (MedMeak), maximum of means (MaxMeak), median of medians (MedMedK) and maximum of medians (MaxMedK), showed *K* = 6 as most likely (Figure 2A). Therefore, results indicated that all six populations had a genetically unique pattern (Figure 2B). At *K* = 2, clear genetic differentiation was observed between SH and all other populations. At *K* = 3, SC began to split from the group, while the rest of the populations remained clustered. At *K* = 4, the six *Z. tyaihyonii* populations could be divided into four groups: the first and second groups consisted of SC and SH alone, respectively, and the third group comprised BS and YC. The last group consisted of CW and TG. Additionally, at *K* = 5, TG was genetically distinct from CW (Figure 2B).

PCoA results are presented in Figure 3. Along Coordinate 1 (11.34%), genotypes from SH were shown to the far left and clustered very distinctly from all other populations, and along Coordinates 2 (7.20%) and 3 (5.2%), genotypes from SC, and BS and YC were clustered distinctly from all other populations, respectively.

To establish phylogenetic relationships among the 94 genotypes, a cladogram was constructed using the UPGMA technique and 3088 SNPs (Figure 4). The tree was clearly divided into geographical populations. Additionally, the tree showed a pattern very similar to those of the STRUCTRUE analysis: SH diverged the earliest, CW and TG, and BS and YC showed the closest relationships.

## 3. Discussion

### 3.1. Genetic Diversity in Z. tyaihyonii Populations

The genetic diversity of species reflects their long-term evolution and adaptation, as well as demographic history [35,36]. High genetic diversity is important because it can allow populations to adapt to environmental changes more easily. Therefore, understanding the level of genetic diversity within and among populations is essential for establishing species conservation strategies [37,38]. To assess the level of genetic diversity in this species, we compared our results with those of related studies that used a method similar to that used in this study. Our comparison showed that *Z. tyaihyonii* has lower genetic diversity (*H_e_* = 0.233) than other species of endemic and woody perennials, such as *Abeliophyllum distichum* (*H_e_* = 0.319) [34], *Tetraena mongolica* (*H_e_* = 0.348) [39] and *Rhododendron rex* (*H_e_* = 0.54) [35]. Given these comparisons, the genetic diversity of *Z. tyaihyonii* can be considered low.

The genetic diversity of plant species is related to their distribution range, population size, life cycle, mating system and gene flow [38,40]. Of these factors, a narrow distribution range and small population size tend to reduce genetic diversity [37,38] as a consequence of genetic drift and inbreeding [14,37,38,41,42,43]. Additionally, in species that reproduce both sexually and vegetatively, such as *Z. tyaihyonii*, the intensity of vegetative reproduction has a significant effect on the genetic diversity of a population. For example, if the intensity of vegetative reproduction is very high, the number of genets becomes extremely low compared to the number of ramets. At the other extreme, if one or only a few genets form a population, the genetic diversity is very low, and even if the population size is large and seed reproduction occurs, the population may face extinction due to high-intensity inbreeding [22,44,45]. The results of this study, however, showed no sign of within-population inbreeding, with all populations displaying a *G_IS_* value close to or even below zero. Therefore, it can be inferred that the reason for the low genetic diversity of *Z. tyaihyonii* is probably the effect of genetic drift. Many limestone (including dolomites) areas are destroyed each year due to mining, and roadsides and mountain edges, which this species prefers, are also areas with very high human interference. For these reasons, many populations of this species, and many individuals within each population, would have disappeared in a relatively short period of time.

In a previous study based on the inter simple sequence repeat (ISSR) method [22], the genetic diversity (Shannon’s index (*I*) = 0.336) of this species was evaluated as relatively high, in contrast to the findings of this study. However, a direct comparison cannot be made because the mean values of genetic parameters may differ depending on the markers used in the analysis [37]. However, since this study was conducted including almost all native populations, we believe that the reliability of our study results will be higher than that of the previous study.

The lowest *He* value among the populations of *Z. tyaihyonii* was observed in SH (*H_e_* = 0.170), followed by SC (*H_e_* = 0.225) and TK (*H_e_* = 0.245). SH is located in the southernmost region of the distribution range of *Z. tyaihyonii* and is the most geographically isolated of all populations (approximately 30 km from the nearest population, YC). Additionally, the size of this population was the smallest (habitat size: approximately 250 m^2^, number of ramets: fewer than 40) among the investigated populations. Additionally, SC is located the farthest north among the *Z. tyaihyonii* populations and was confirmed to have the second smallest population size (after SH). Therefore, the low genetic diversity in these populations is thought to be due to their small size and repeated founder events [46,47,48].

### 3.2. Genetic Differentiation and Population Structure

Two important parameters, gene flow (*N_m_*) and the genetic differentiation coefficient (*F_ST_*), are employed to assess the genetic structure of populations, and they are negatively correlated [14,49,50]. Our data also showed this trend (Appendix A). According to Wright [51], *Nm* is divided into three grades, high (≥1.0), medium (0.250–0.99) and low (0.0–0.249), and when *Nm* > 1 there is gene flow between populations [39,52]. Additionally, Wright [51] proposed that if the *F_ST_* value of the populations is between 0 and 0.05, there is no genetic differentiation among populations; if the *F_ST_* value is between 0.05 and 0.15, they are moderately differentiated; if the *F_ST_* value is between 0.15 and 0.25, they are highly differentiated. The results of this study suggested that there was a high level of gene flow between all pairs of populations except those including SH. Additionally, the results showed a moderate level of genetic differentiation between BS, CW, TG and YC, and a high level between a population and SH or SC. Flowers of *Z. tyaihyonii* are entomophilous (pollinated by insects), and fruits with elongated calyx lobes are dispersed by wind [53,54]. Since the fruits are relatively large (approximately 5 mm) and cannot spread far [22], the high gene flow detected in this study is probably explained by insect pollination. Unfortunately, there have been no studies on the pollinators of this species, so further research is needed.

The AMOVA results (*p* < 0.001) supported a high level of population differentiation and showed major molecular variance within individuals rather than among populations. In outcrossing and long-lived plants in general, most genetic variation exists within populations, while selfing plants maintain the majority of genetic variation among populations [55]. Therefore, these results suggest that this species maintains its populations mainly by seed propagation.

Gardner and Mangel [56] predicted that favorable habitats would promote clonal growth over sexual reproduction. Additionally, Zhang and Zhang [57] reported that most resources should be allocated to sexual reproduction in habitats with changing environmental conditions and intense competition, whereas clonal growth should be dominant in stable habitats. *Z. tyaihyonii* grows naturally in only a very limited portion of the dolomitic limestone areas in the Republic of Korea [4,7,8,9,10,23]. Thus, this species appears to prefer the environmental characteristics offered by dolomitic limestone areas rather than being able to grow only in dolomite areas. Dolomite soils are rich in Mg and poor in nutrients, and when tectonic or weathering processes generate skeletal soils, predominantly sandy or gravelly in texture, they further complicate water retention, and as a result, make them very dry [58,59,60]. They differ chemically from their non-carbonate counterparts primarily in that they have a higher pH, and lower Fe, P and K [61]. Additionally, the dolomite distribution area is characterized by undeveloped crevices, and the thickness of the soil layer is generally less than 20 cm, which is unsuitable for the distribution of woody plants with deep root [62]. This probably provided a very suitable environment for the settlement of high light-demanding plants such as *Z. tyaihyonii*. However, the development of such vegetation would have made it easy for nutrients to accumulate in the soil, which would have facilitated the influx of competing plants, including subtrees and trees. Because of the consequences from these processes, this species is believed to maintain populations by seed propagation rather than by vegetative propagation.

A variety of methods are used to detect genetic diversity and population structure [63,64,65,66,67], and Wang [50] suggested combining three effective techniques, PCoA, STRUCTURE analysis and the UPGMA technique, to obtain reliable results. The STRUCTURE analysis in this study optimally divided the six selected populations into six clusters (Figure 2). This pattern was strongly supported by the UPGMA (Figure 4) result. These results will be useful in identifying important management units for effective conservation strategies.

### 3.3. Implications for the Conservation of Z. tyaihyonii

In this study, we evaluated the level and degree of genetic variation within and among populations, as well as the connectedness among populations of *Z. tyaihyonii* to provide useful information for the effective conservation planning of this endemic and rare plant. Genetic diversity is especially important for preserving the latent evolutionary capacity of a species to deal with changing environments [68]. In addition, information about genetic variation within and among populations in endangered and rare plants plays an important role in the process of formulating conservation and management strategies [69]. *Z. tyaihyonii* has a very low evolutionary potential to adapt to changing environmental conditions due to its low genetic diversity. Additionally, the results of this study suggested that inbreeding does not represent an immediate danger for this species, but genetic drift would have had a significant impact on population decline. The greatest reason for the low genetic diversity of this species is believed to be bottlenecks caused by the direct destruction of habitats by human interference, such as limestone quarries including dolomites, related facility construction, road construction, and cultivation activities. Additionally, the interspecies competition that accompanies the vegetation succession process was also considered to have a significant impact. For this species to be conserved naturally in situ, therefore, priority should be given to directly removing competing plants and minimizing disturbance by humans.

Protection of trailing-edge populations should be prioritized because the prolonged isolation and successful persistence of such groups suggest their evolutionary potential, which can be useful for the long-term conservation of genetic diversity and phylogenetic history of the species [34,70]. Additionally, isolated populations may accumulate through selective and stochastic processes, the genetic and chromosomal differences that render them independent and distinctive evolutionary entities, and if speciation involves few major changes promulgated by selection and/or genetic drift in small populations, speciation may be relatively rapid, perhaps saltational [71,72]. SH is believed to be more important than other populations, as it is located at the southernmost limit of the distribution range of this species. This population was revealed as the most heterogeneous and isolated by various results from this study, such as genetic differentiation, analysis, barrier reconstruction, STRUCTURE analysis, PCoA and the UPGMA technique. These findings suggest that transplanting individuals from SH into other populations and vice versa is not an advisable conservation strategy for this species. SH has a very high risk of natural culling due to its very small size (fewer than 40 individuals) and a very low genetic diversity (*He* = 0.170). Therefore, more active efforts to protect this population will be needed, such as setting up conservation areas and conducting periodic monitoring.

Given its geographic location at the northernmost limit of the distribution range of the species, the SC population is not recommended as a source of transplants, as previously suggested for SH. SC showed high gene flow with other paired populations except for SH and showed moderate genetic differentiation from BS (Appendix A). This probably means that pollen is transported by pollinators between this population and other populations except for SH (at least between this population and BS).

A recommended conservation strategy for endangered plants, in addition to in situ conservation, is the establishment of ex situ gene banks for each population in both the field and laboratory. Although this species is not listed as an endangered wild plant under legal protection in Korea [73], it is a serious challenge that most of its populations are adjacent to limestone quarries, including dolomites, villages, and roadsides, and are prone to destruction. Given this current situation, we believe that ex situ conservation of this species should be considered. For ex situ conservation of this species, conservation measures that include all populations with the goal of maximizing the collection of private alleles in each population would be most appropriate because the STRUCTIRE and UPGMA results showed that each population formed an independent clade, and the genetic differentiation between populations was relatively high (Figure 4 and Appendix A). Otherwise, ex situ conservation was observed by targeting private alleles of the four genetic groups (the first and second groups were SC and SH alone, respectively; the third group comprised BS and YC; and the last group consisted of CW and TG) identified in the UPGMA analysis (Figure 4).

## 4. Materials and Methods

### 4.1. Plant Materials

Since *Z. tyaihyonii* is not an endangered or protected species, plant materials were collected without permission. Distribution information for sampling was confirmed through the literature and specimens from the National Institute of Biological Resources and the Korea National Arboretum. A total of 94 individual plants were collected from six natural populations, and 14 to 22 individuals in each population were sampled at least 5 m apart to minimize the likelihood of sampling clones. During field sampling, the sampled site and altitude were recorded (Figure 1 and Appendix A). One fresh leaf per individual was placed in plastic bags and placed on ice to be transported to the laboratory as soon as possible, where they were stored at −20 °C until DNA extraction. The voucher specimens for each population were deposited in the Sangji University Herbarium (SJUH).

### 4.2. DNA Extraction, Library Preparation, and Sequencing

Total genomic DNA was extracted using the DNeasy Plant Mini Kit (Qiagen Inc., Valencia, CA, USA), following the instructions of the manufacturer. The quality and quantity of extracted DNA were evaluated with a LabChip GX II (PerkinElmer Inc., Waltham, MA, USA), and DNA was visualized on a 1% agarose gel. Samples that passed quality and quantity filters were subsequently sent to Xenotype Inc. (Daejeon, Republic of Korea) for GBS library construction [27]. Using this approach, samples were digested with methylation-sensitive *ApeKI* restriction enzyme and barcoded with sequence adapters before pooling into sets of 94 individuals. Each library was amplified by PCR and sequenced in a single lane on a HiSeq X instrument (Illumina Inc., San Diego, CA, USA) using 150 bp paired-end sequencing runs.

### 4.3. Sequence Analyses, Bioinformatics, and SNP Identification

NGSEP Wizard is a tool that implements all the steps necessary to obtain a population variant file with just one run screen, starting from raw sequencing data [74]. We obtained the VCFfile of SNPs using NGSEP v4.2.1. SNPs were removed using VCFtools v.4.2 [75] on the basis of the following filtering criteria: SNP missing data ≥ 50%, minor allele count ≥ 3, minimum quality score < 30, minimum mean depth ≤ 3, individuals missing data ≤ 50%, SNP data < 95%, minor allele frequency < 5%, minimum mean depth ≤ 20, and *p* value for Hardy–Weinberg equilibrium ≤ 0.01. Then, linkage disequilibrium (LD) (*r^2^* > 0.8) was computed using the Plink v. 1.07 software [76]. The online program PGDSpider v.2.1.1.5 [77] was used to convert different file formats for all downstream analyses described below.

### 4.4. Data Analyses

The number of alleles (*Na*), the number of effective alleles (*Ne*), Shannon’s information index (*I*), observed heterozygosity (*Ho*), expected heterozygosity (*He*), AMOVA, gene flow (*Nm*), and PCoA were computed with the GenAlEx 6.5 software [78,79]. PCoA was visualized 3-dimensionally using the R package Scatterplot3d [80], and the inbreeding coefficient (*G_IS_*) was calculated by Genodive v.2.0b27 [81]. Population genetic structure was analyzed using a Bayesian clustering analysis method conducted in STRUCTURE v2.3.4 software [82]. An admixture model was used to estimate the number of population clusters (*K*) ranging from one to six, and each STRUCTURE run was performed with a burn-in size of 50,000 and 100,000 MCMC iterations with 10 runs per *K* value. Results were fed into StructureSelector [83] to identify the most likely number of clusters present based on the MedMeak, MaxMeak, MedMedK and MaxMedK criteria [84]. The results of population clustering were aligned and visualized using the CLUMPAK webserver [85]. The genetic relatedness of individuals was calculated using poppr [86] and adegenet [87] in R 4.0.4 [88]; then, a cladogram was constructed based on the UPGMA algorithm in R 4.0.4 [88]. Monmonier’s maximum difference algorithm was used with BARRIER v. 2.2 [89] to explore the geographical sites exhibiting maximal genetic discontinuities among populations. Sampling sites were mapped based upon geographical coordinates using this program, with barriers being indicated on the map by assessing maximum values within a population-pairwise genetic distance matrix.

## 5. Conclusions

A total of 3088 high-density SNPs were employed to estimate the genetic diversity and population structure in 95 *Z. tyaihyonii* accessions from six populations. This species maintained low genetic diversity levels, high genetic differentiation, and high gene flow among populations except for SH. This pattern mainly results from human interference, which gives this species a very low evolutionary potential to adapt to changing environmental conditions. For this species to be conserved naturally in situ, directly removing competing plants and minimizing disturbance by humans are very important. Additionally, special management of SH, which is highly isolated geographically and genetically, should also be included. This is the first study of the genetic diversity and population structure of *Z. tyaihyonii* using genotype-by-sequencing (GBS). The results of this study and our suggestions for conservation will serve as important information for the population conservation and management of this species.

## Figures and Tables

**Figure 1 plants-12-00171-f001:**
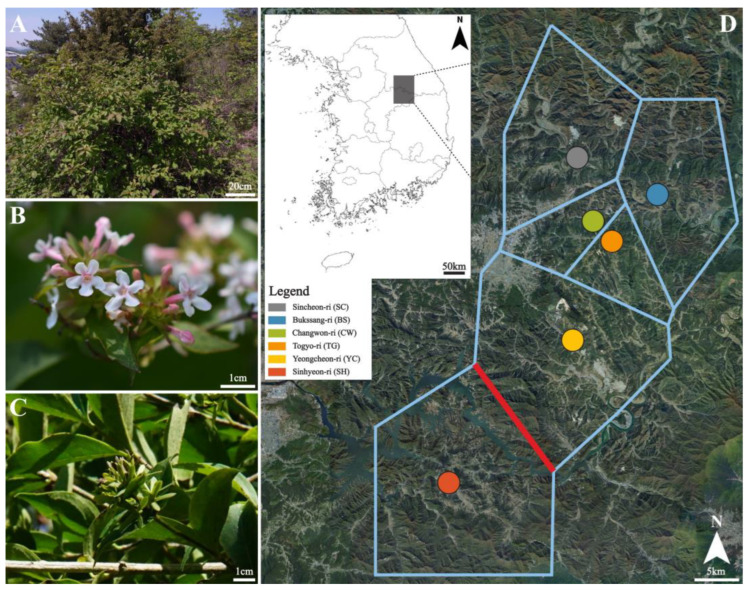
Photos of habitat (**A**), flowers (**B**) and fruits (**C**), of Taihyun’s abelia (*Zabelia tyaihyonii*), and (**D**) geographic distribution of six populations of *Zabelia tyaihyonii*, and genetic barriers (red lines) computed from pairwise *F_ST_*. Gray lines correspond to hypothetical boundaries between populations, which are labeled with corresponding codes.

**Figure 2 plants-12-00171-f002:**
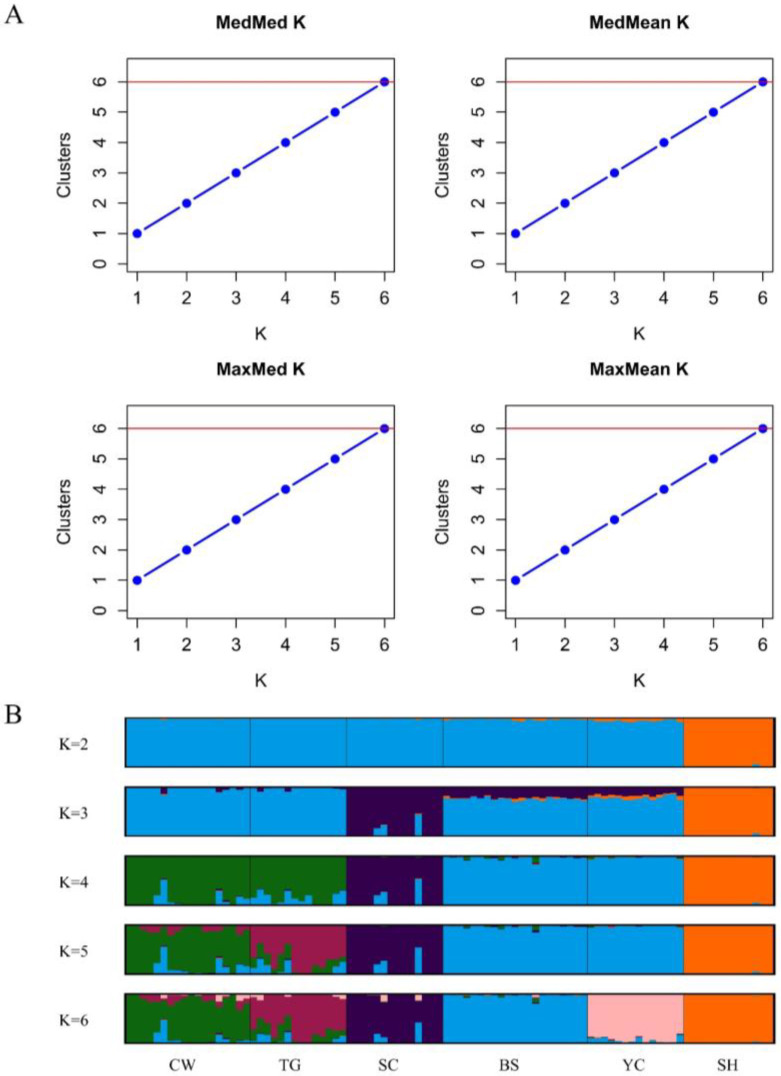
Population structure of 94 *Z. tyaihyonii* individuals from 6 different populations. (**A**) The optimal *K* number indicated by alternative measures MedMeak, MaxMeak, MedMedK and MaxMedK applied in StructureSelector and (**B**) STRUCTURE bar plot (*K* from 2 to 6). Each bar represents a single individual and its proportion of assignment to each of the sample sites separated by solid black lines.

**Figure 3 plants-12-00171-f003:**
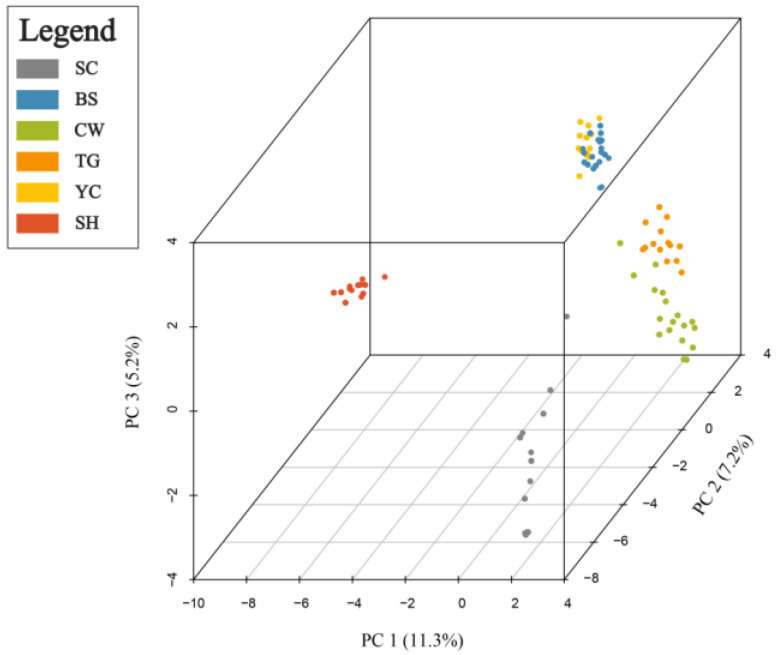
Results of principal coordinate analysis (PCoA) of genetic distances for all 94 genotyped individuals.

**Figure 4 plants-12-00171-f004:**
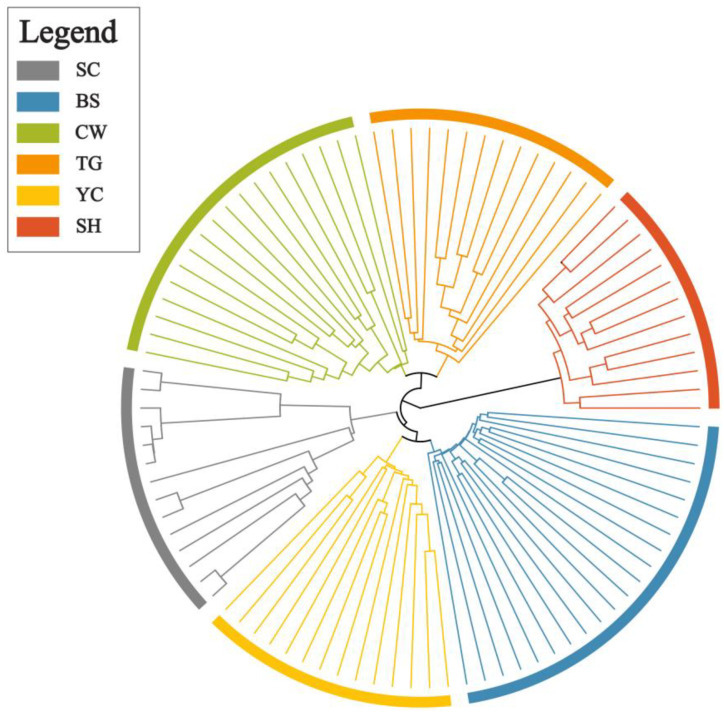
A cladogram of the 94 *Zabelia tyaihyonii* accessions from six populations. Different colors represent different populations.

**Table 1 plants-12-00171-t001:** Estimates of genetic diversity within *Zabelia tyaihyonii* populations.

Pop.	N	Na (±SE)	Ne (±SE)	I (±SE)	Ho (±SE)	He (±SE)	G_IS_
**BS**	21	1.881 (±0.006)	1.428 (±0.006)	0.406 (±0.004)	0.266 (±0.003)	0.263 (±0.003)	0.014
**CW**	18	1.821 (±0.007)	1.419 (±0.006)	0.384 (±0.004)	0.262 (±0.004)	0.252 (±0.003)	−0.013
**TG**	14	1.797 (±0.007)	1.405 (±0.006)	0.375 (±0.004)	0.273 (±0.004)	0.245 (±0.003)	−0.079
**SC**	14	1.671 (±0.008)	1.380 (±0.007)	0.339 (±0.005)	0.267 (±0.005)	0.225 (±0.003)	−0.150
**YC**	14	1.791 (±0.007)	1.405 (±0.006)	0.375 (±0.004)	0.260 (±0.004)	0.245 (±0.003)	−0.025
**SH**	13	1.441 (±0.009)	1.299 (±0.007)	0.249 (±0.005)	0.209 (±0.005)	0.170 (±0.004)	−0.194
**mean**		1.734 (±0.003)	1.389 (±0.003)	0.355 (±0.002)	0.256 (±0.002)	0.233 (±0.001)	−0.057

N, no. of samples; Na, no. of alleles; Ne, no. of effective alleles; I, Shannon’s information index; Ho, observed heterozygosity; He, expected heterozygosity; G_IS_, inbreeding coefficient; SE, corresponding standard error.

**Table 2 plants-12-00171-t002:** Results of analysis of molecular variance (AMOVA) and significance after 20,000 permutations.

Source	Df	SS	MS	Est. Var.	%	F-Statistics	*p* Value	Nm
Among populations	5	14,276.121	2855.224	79.490	17.00	*F*_ST_ = 0.170	*p* < 0.001	1.221
Among individuals	88	33,547.943	381.227	−6.993	−1.5	*F*_IS_ = −0.018	*p* > 0.05	
Within individuals	94	37,150.0	395.213	395.213	84.50	*F*_IT_ = 0.155	*p* < 0.001	
Total	187	84,974.064		467.71	100.00			

Df, degrees of freedom; SS, sum of squares; MS, mean of squares; Est. Var., estimated variance; %, percentage of variation; Nm, inferred gene flow.

## Data Availability

Not applicable.

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
