# Peer review of "Population Genomics Study and Implications for the Conservation of Zabelia tyaihyonii Based on Genotyping-By-Sequencing"

_plants, 2022, doi:10.3390/plants12010171_

Round 1

Reviewer 1 Report

Dear Authors

Reviewer report:

Regarding the manuscript entitled “Population genomics study and implications for the conservation of Zabelia tyaihyonii, an endemic and rare plant in Korea, based on genotyping-by-sequencing (GBS)” with ID plants-2071074

The present study aimed to clarify the genetic diversity and population structure of Zabelia tyaihyonii, an endemic species to South Korea with implications in conservation and management. The topic is interesting and has valuable information. The title is soo long and it should be revised and corrected to be sound. However, the most serious issue in the article is the taxonomic identification of the plant. In some cases, the plant name was written wrong, please revise the WFO; http://www.worldfloraonline.org/taxon/wfo-0000430178. Also, the family is not right as “Linnaeaceae” this must be revised as Caprifoliaceae. It was mentioned that “It was described as a new species on the basis of two morphological differences from Zabelia mosanensis” although this plant “Abelia mosanensis” is a synonym of Zabelia tyaihyonii. The authors should revise all synonyms of this plant [Abelia mosanensis T.H.Chung ex Nakai, Abelia tyaihyoni Nakai, and Zabelia mosanensis (Chung ex Nakai) Hisauti & Hara)]. The results are well presented but the figures need adjustment. The discussion is well written with comparison and sophistication with the present and published data, however, I suggest providing a conclusion as a separate section. Supplementary materials need revision and coupling of the figures and tables in the same file. Therefore, I recommend a major revision of the article.

Sincerely Yours,

I proposed some suggestions that could improve the manuscript

Title

·        The title is long, it should be revised to be sound.

·        No need for abbreviations in the title, please delete the abbreviation.

Abstract

·        Line 23-25: “STRUCTURE analysis optimally divided the six studied populations into two clusters, with similar results obtained using principal coordinate analysis (PCoA) and the unweighted pair group method with arithmetic mean (UPGMA) technique.” this must be shifted up with materials and methods, and here add the obtained results of these analyses.

·        The abstract should have a solid conclusion based on the results.

Keywords:

·        The keywords should be corrected without repetition from the title.

Introduction

·        Line 32: the introduction needs to be revised and not started with talking about soil. You should go with the main topic of the article.

·        Line 42: “Linnaeaceae” this is wrong! Replace with “Caprifoliaceae”.

·        Line 44: “Zabelia mosanensis (T. H. Chung ex Nakai):” this is not right “Abelia mosanensis T.H.Chung ex Nakai”.

·        Line 34-44: “It was described as a new species on the basis of two morphological differences from Zabelia mosanensis” although this plant “Abelia mosanensis” is a synonym of Zabelia tyaihyonii. Revise!!!!!

·        Line 47-49: revise this section.

Results,

·        Figure 1:

1.      What about the polygon? And the Genetic barrier? How it was performed? No data in materials and methods regarding this analysis.

2.      I suggest providing an overview of the whole plant and adding a scale bar for the photos.

·        Figure 2:

1.      The A & B has very low resolution, please adjust.

2.      The legend should not start with (A)!!! revise.

·        Figure 3: PC 1 & 2 has low values of 11.34% and 7.20!! You should revise and try another PC, maybe PC3.

·        Figure 4: what is the scale 0.05? should be adjusted abd clarified in the legend.

Discussion,

·        Is well written with comparison and sophistication with the present and published data.

·        I suggest providing a conclusion as a separate section.

Materials and Methods

·        Line 303-310: The sample collection should be adjusted. What part was collected? How it was collected? What quantity? Collected in what? How did it transfer? …..etc.

·        Line 307: change fourteen to 14.

·        Line 309-310: “The voucher specimens for each population were deposited in the Sangji University Herbarium (SJUH).” The voucher authentication should be provided.

Supplementary materials

·        Figure S1 should be coupled within one file with Table S1.

·        The legend of Figure S1 must be provided.

·        The coordinates of Table S1 are suggested to be in UTM format.

References

The References are formatted well.

Reviewer 2 Report

Kang et al. conducted a population genomics analysis of Zabelia tyaihyonii natural populations based on SNP data generated from genotyping-by-sequencing (GBS). The authors report on ‘low’ genetic diversity and no sign of inbreeding, and a high degree of genetic differentiation among sampling populations. Their findings provide novel insight for the delieating of conservation units with implications on genome-informed provenance-sourcing of Z. tyaihyonii.

I found that the manuscript was well-written, and the discussion section was robust. However, the analytical workflow of the paper could be improved to obtain robust results to support the number of genetic clusters and quantifying contemporary gene flow rates.

Lines 333-346 – 4.4 Data analysis: It is not noted on how gene flow (Nm) was calculated here. Anyways, the appropriate approach to quantify contemporary migration is by using BayesAss3-SNPs (BA3-SNPS, Mussmann et al. 2019 - Methods in Ecology and Evolution - https://doi.org/10.1111/2041-210X.13252). Likewise, the method used for determining the optimum number of genetic clusters is no longer the standard in the field. To infer the ideal K, the field has advanced to adopting the Puechmaille technique, which is based on four estimators appropriate for unevenly sampled population: (1) median of means, MedMeaK; (2) maximum of means, MaxMeaK; (3) median of medians, MedMedK; and (4) maximum of medians, MaxMedK (Puechmaille et al. 2016: https://doi.org/10.1111/1755-0998.12512) as implemented on the STRUCTURESELECTOR web server (https://lmme.ac.cn/StructureSelector/?_ga=2.205138200.988624382.1669979777-1616063975.1669979777).  Additionally, one should use of EvalAdmix (https://doi.org/10.1111/1755-0998.13171; http://www.popgen.dk/software/index.php/EvalAdmix) for evaluating model fit or the ‘appropriateness’ of the inferred K values from STRUCTURE. Likewise, Wang (2019; https://doi.org/10.1111/1755-0998.13000) previously proposed the use of the parsimony index (PI) implemented in a computer program, KFinder. The Puechmaille and EvalAdmix approaches are the two methods I would recommend for the authors to conduct and present in their manuscript.

I recommend revisions. 

Reviewer 3 Report

Abstract

I think it is not necessary to repeat in the abstract terms or strings of words that are already covered in the title. This repetition does not make bibliographic searches more efficient for other authors who may be interested in the article. The term "rare plant" is contemplated both in the title of the article and in the abstract.

The abstract must adapt to this structure:

1. indicate main objectives and scope of the investigation,

2. detail the methods used,

3. summarize the results, and

4. state the main conclusions

The objectives are not specified, moreover, as far as the conclusions are concerned, the abstract also seems incomplete to me.

Introduction

The introduction is poorly structured. In an article that deals with the genomics of a species and its conservation, one cannot begin by talking about the edaphic characteristics of the soils in which it lives. First of all, you have to talk about the species, its degree of threat and then deal with its habitat. In the latter case, it is necessary to describe it well and try to expose the causes that may have led to the fact that not only this species is linked to the soil characteristics of the environment. The described crumb structure and its drainage reminded me a lot of what is described in https://doi.org/10.1016/j.flora.2007.06.006. See next comment.

Lines 32-40.- For me it is important that it is clear if it is limestone or dolomites (also called xeric-limestone). Frequently the high percentages of endemicity do not occur on limestone, but on dolomite (cf. https://doi.org/10.3390/biology10010038). Other aspects that are mentioned when describing the habitat of Zabelia tyaihyonii are very reminiscent of the so-called “dolomitophily” or “edaphism” phenomenon. According to some authors, this phenomenon is globally widespread.

Lines 47-49.- It is said that Zabelia tyaihyonii is threatened, but later (lines 289-290) it is said otherwise or so it may seem. This circumstance may be due to the fact that the species does appear on the red lists, but it is not legally or effectively protected. If so, this dissonant aspect should be underlined throughout the manuscript. The discrepancy between the red lists versus nature protection Acts has already been highlighted by different authors. I think it is a key issue in the manuscript on which little insistence is made.

Lines 53-54.- The statement made here is very strong. I have not been able to access the source that is mentioned. Maybe because it's written in Korean? In my opinion, the authors should indicate which publications are originally written in this language, although perhaps this is a requirement that the editors of the journal should make. In any case, I don't think it can be generalized that “endemic plants are very sensitive to subtle changes in environmental factors”. I would like this claim to be backed up with some references.

Fig. 1.- What is the argument for considering the CW and TG populations as different? What geographical distance separates them? The relationship between the geographic distances of populations and their genetic distance is perhaps worth noting.

However, the most important shortcoming of the introduction is that there is no starting hypothesis or clearly specified objectives. This problem, as already mentioned, also appears in the abstract.

Discussion

Lines 156-158.- If the data referring to its area of occupation and number of ramets is available for each population, wouldn't it be interesting to reflect it in table 1 or in another table?

Lines 231-234.- I insist, to clarify aspects of the ecological requirements of this plant, first it is necessary to clarify if it lives on dolomites and then other aspects related to these soils if, as I suspect, this plant is strictly dolomites or tolerant to dolomites This type of rock is sometimes compact or may be finely crushed, which increases its xericity.

Line 269.- What kind of quarrying? What minerals or materials are extracted from there? Marble, gravel?

Lines 282-283.- I don't quite understand why the SC population is not considered of interest for transplants. Perhaps because it already naturally interbreeds with almost all other populations? In that case, it might be interesting if those other populations need to be reinforced at some point. Please discuss this point.

289-291.- Is this species not included in any protected area? There is no mention at any time (or almost) of in situ conservation in Korea and its protected areas.

References

Line 371.- Jeffery should be Jeffrey

Round 2

Reviewer 1 Report

Dear Authors,

I appreciate your revision and consideration of my comments and suggestions. I recommend publication of your article.

Sincerely 

Author Response

I appreciate your revision and consideration of my comments and suggestions. I recommend publication of your article.

→ Thank you very much for recommending that our article be published. Your comments in the first review have been very helpful in improving our thesis academically. Thank you very much again.

Reviewer 2 Report

Excellent revision work. I believe it is best to highlight K2 to K6 so that the observed pattern of hierarchical population structure, which corresponds to the hierarchical clustering depicted in Figure 4, is not completely overlooked. Furthermore, I would keep the deleted text in Lines 164-169 and expand on it by describing the clustering patterns at K5 and K6. These findings also shed light on the phylogeography of the studied species.

Author Response

Excellent revision work. I believe it is best to highlight K2 to K6 so that the observed pattern of hierarchical population structure, which corresponds to the hierarchical clustering depicted in Figure 4, is not completely overlooked. Furthermore, I would keep the deleted text in Lines 164-169 and expand on it by describing the clustering patterns at K5 and K6. These findings also shed light on the phylogeography of the studied species.

→ Thank you very much for your nice comments. We accepted your comments and presented the STRUCTURE bar plots at K from 2 to 6 in Figure 2. Also, we have changed some sentenses related to it.